# Microbiomes, Their Function, and Cancer: How Metatranscriptomics Can Close the Knowledge Gap

**DOI:** 10.3390/ijms241813786

**Published:** 2023-09-07

**Authors:** Lina Aitmanaitė, Karolis Širmonaitis, Giancarlo Russo

**Affiliations:** EMBL Partnership Institute for Gene Editing, Life Sciences Center, Vilnius University, LT-10257 Vilnius, Lithuania; lina.aitmanaite@gf.vu.lt (L.A.); karolis.sirmonaitis@gmc.vu.lt (K.Š.)

**Keywords:** metatranscriptomics, metagenomics, cancer, microbiome

## Abstract

The interaction between the microbial communities in the human body and the onset and progression of cancer has not been investigated until recently. The vast majority of the metagenomics research in this area has concentrated on the composition of microbiomes, attempting to link the overabundance or depletion of certain microorganisms to cancer proliferation, metastatic behaviour, and its resistance to therapies. However, studies elucidating the functional implications of the microbiome activity in cancer patients are still scarce; in particular, there is an overwhelming lack of studies assessing such implications directly, through analysis of the transcriptome of the bacterial community. This review summarises the contributions of metagenomics and metatranscriptomics to the knowledge of the microbial environment associated with several cancers; most importantly, it highlights all the advantages that metatranscriptomics has over metagenomics and suggests how such an approach can be leveraged to advance the knowledge of the cancer bacterial environment.

## 1. Introduction

The symbiotic relationship between humans and the plethora of bacteria inhabiting the human body has gained much-deserved attention in the past twenty years [1,2] and it is now evident that the various consortia of microorganisms present in our body have direct consequences on our health [3]. In particular, the presence/absence of certain microbes [4], a lack of balance in the overall species’ abundances [5], their specific functional capabilities [6], and shifts in the downstream metabolic pathways [7] are all factors that can favour the onset or the progression of a number of diseases. Cancer is no exception: alterations of the microbiome have been linked to breast cancer [8], cervical cancer [9], colorectal cancer (CRC) [10], gastric cancer [11], oral carcinoma [12], lung cancer [13], melanoma [14], and others [15]. 

The advent of -*omics* technologies has enabled the surveying of entire molecular landscapes (e.g., the whole transcriptome) in cells [16,17,18], advancing the knowledge of how the combined make-up of several genes or the concerted activities of several proteins can affect more macroscopic phenotypes [19]. The application of such approaches to communities of organisms spurred *meta-omics* experiments, i.e., experiments whose goal is to characterise in one or several ways a bacterial community. *Meta-omics* experiments can be broadly categorised into four groups: 

(1) 16S rRNA metagenomics amplifies and sequences the ribosomal 16S RNA gene, a small section of the bacterial genome (about 1.6 Kb) which contains a mixture of conserved and variable sequences [20]. The conserved parts ensure that the majority of the bacteria are represented, while the variable parts allow one to distinguish between microorganisms and therefore determine their relative abundances and the diversity of the community.

(2) In a shotgun metagenomics (MGx) experiment [21], one extracts, shears, and sequences whole genomes from bacterial cells in the community. Therefore, through the production of full genome assemblies, this is the method that is most capable of accurately locating a genome in a phylogenetic tree and thus identifying novel species. However, the functional potential of the members of the community can be only predicted.

(3) Metatranscriptomics (MTx) [22] sequences the whole transcriptomes of the bacterial cells in a community, thus directly inferring the function and enabling the quantification of the expression of the transcribed genes.

(4) Metaproteomics [23] captures and quantifies the active proteins in a community. A summary of the complete workflows for these approaches is shown in Figure 1. 

The purpose of this review is to highlight the major contributions of metagenomics and metatranscriptomics in the field of cancer research and especially to remark on the added value of metatranscriptomics relative to 16S rRNA and shotgun metagenomics; we also review the interaction between the microbial communities and transcriptome of the host, sometimes also referred to as metatranscriptomics; finally, we summarise some general guidelines to help exploit the technology to the full.

## 2. Summary of the Studies Reviewed 

A breakdown of the studies included in this review, split by methodology, is shown in Table 1. The majority of the studies (56%) have used 16S rRNA alone, while MTx alone and MGx alone each represent a little less than 20% of the total. In the past, 16S rRNA was often performed alongside MGx and used to establish a “ground truth” regarding the composition of the microbial community. With improvements in the algorithms devoted to analysing shotgun data, this is not necessary any longer, and in fact only three studies (2.1%) combined 16S rRNA with MGx. Lung cancer is analysed in most papers (18), immediately followed by CRC, breast cancer, and oral carcinoma, which together represent 52% of the total number of papers analysed. This is not surprising, considering that the gut, oral, and recently the airways floras are the most studied human microbiomes.

Across the studies reviewed, some findings have been replicated, and they constitute the most consistent evidence on the interaction between the various human microbiomes and cancer. These findings, summarised in Figure 2, include enrichment of specific species and pathways, difference in the diversity of the microbial community, and relocation of commensal species. 

## 3. Breast Cancer

Breast cancer (BC) is the fourth deadliest cancer overall and every year represents over 40% of all new cancer diagnosed among women [24]. The correlation between human microbiomes and BC has not been investigated until recently. Three microbiomes have been analysed in association with BC: from breast cancer tissue (intraoperatively or aspirate), breast-adjacent tissue, and faeces. The majority of the research associating microbiomes to BC is based on 16S rRNA sequencing, while only a few studies analysed the metatranscriptome, either sequencing it from scratch or using already available data, such as data from The Cancer Genome Atlas Program (TCGA) [25]. In some cases, the virulence genes of the microbiome were identified using PathoChip, a gene array comprising more than 3700 probes from almost 1400 species, covering over 7400 genes [26]. 

Although certain species have been consistently reported as enriched or depleted in cancer samples, there is also inconsistent evidence at this regard (Figure 3) and breast cancer is no exception, especially when comparing it to normal adjacent tissue and healthy tissue. Most of the studies report higher diversity and richness in BC than in healthy tissue [27,28,29]. In addition, there is no clear consensus on whether tumour and tumour-adjacent tissue differ significantly, as some studies did not find any differences [28,30,31]. In spite of this variability, multiple studies could reproduce the enrichment of certain bacteria in breast cancer patients. As an example, phyla *Firmicutes* [29,32], *Fusobacteria* [28,32,33,34], and *Bacteroidetes* [30,32,34]; families *Alcaligenaceae* [27,35], *Streptococcaceae* [8,32,34], *Enterobacteriaceae* [8,28,30], *Pseudomonadaceae* [28,29,32], *Rhodobacteraceae* [27,36], *Propionibacteriaceae* [35,36], and *Micrococcaceae* [27,36]; and genera *Bacillus* [8,27,30] and *Staphylococcus* [30,35] are among them.

In contrast, opposite results have also been reported, such as the enrichment of genera *Corynebacterium*, *Prevotella*, *Streptococcus*, *Micrococcus*, *Propionibacterium*, *Staphylococcus*, and *Lactococcus* in healthy tissues and a decrease in the relative abundance of family *Bacteroidaceae* with cancer development [28,30,36,37]. There has been evidence of both *Methylobacterium* enrichment and depletion in cancerous tissues [35,38], and many more microbes have been associated with breast cancer in single studies. PathoChip enabled the identification of different microbial, viral, and fungal profiles of breast cancer, but of course, its power is limited by the intrinsic bias of probe-based systems [26]. It was also found that the microbial load in breast cancer tumours is the highest compared to other cancers [39]; however, in general, tumours seem to have a lower bacterial load than healthy tissues, and it further decreases with progression [40]. 

In cancerous tissue, functional pathway predictions using PICRUSt revealed upregulation of pathways related to glycerophospholipid metabolism, ribosome biogenesis, and anaerobic respiration, while pathways related to inositol phosphate and flavonoid metabolism were downregulated [33,36,39]. 

A number of studies investigated how the functioning of the host molecular machinery (e.g., gene expression or protein levels, pathways activation, and immune response) changed depending on the composition of the microbiomes. For example, a higher abundance of *Listeria* spp. has been found to be associated with the expression of genes involved in epithelial-to-mesenchymal transition (EMT), while that of *Haemophilus influenzae* was associated with proliferative pathways, including G2M checkpoints and E2F transcription factors [41]. In another study based on TCGA data, Parida et al. [25] reported that environmental-information-processing pathways, oncogenic pathways, and lipid metabolism pathways are significantly enriched in tumour samples. In particular, phosphatidylinositol signalling, mTOR signalling, calcium signalling, ABC transporters, and phosphotransferase system were associated with the ethnicity of the patient, being upregulated in the tumours of black women but not in those of white or Asian women. By integrating lipidomic and metagenomic data, Giallourou et al. [37] identified lower ceramide and diacylglycerol levels in tissues from breast cancer patients; in particular, the depletion of ceramide was mediated by *Gammaproteobacteria* and *Bacillus* bacteria. In healthy controls, *Acinetobacter*, *Lactococcus*, *Corynebacterium*, *Prevotella*, *Streptococcus*, and *Lactococcus* had a positive correlation with diacylglycerols.

By combining Nanostring-based gene expression and a Cytokine/Chemokine array, Tzeng et al. [28] profiled the immunological landscape of breast cancer patients and determined that cancerous tissue is enriched in T-cells, CD8+ T-cells, natural killer cells (NKC), neutrophils, and FOXP3+, and depleted in the number of dendritic cells and macrophages. Moreover, tissue toll-like receptor (TLR) pathways, which are responsible for sensing microbes, significantly changed in tumour samples through the downregulation of TLR4 and the upregulation of MYD88 and IRAK1. A network analysis of associations between microbiome and immune-related gene expression and cytokine concentrations revealed that *Methylibium*, *Pelomonas*, and *Propionibacterium* were likely mediating those alterations in cellular abundance and gene expression.

## 4. Cervical Cancer

Almost the entirety of cervical cancers are caused by long-lasting Human Papilloma Virus (HPV) infection, and some evidence suggests that the microbiome of the cervical cavity plays a role in the process of infection and clearance [42]. Therefore, HPV-positive patients are often part of the microbiome analysis of cervical cancer cohorts. Such cohorts also include patients with precancerous states, such as low- or high-grade cervical intraepithelial neoplasia. Most cervical cancer microbiome studies used vaginal swab samples, cervicovaginal lavages, or cell biopsies. 

Researchers have repeatedly found that the overall abundance of *Lactobacillus* decreases as non-cancerous lesions progress to cancer [43,44,45]. Such depletion of *Lactobacilli* is often associated with other microorganisms such as *Sneathia* spp., *Megasphaera*, *Shuttleworthia*, *Prevotella*, *Streptococcus*, *Porphyromonas*, and *Fusobacterium* spp. proliferating in precancerous or cancerous environments, [43,44,45,46,47,48]. An increase in immune mediators in cancerous tissue was also detected, and one study claims that this could be associated with the high abundance of *Fusobacterium* spp. in cancer patients [43,44]. *Gardnerella* is another bacterium associated with cervical cancer and HPV infection, although the evidence seems contrasting. On the one hand, some research mainly found it in HPV-negative samples and negatively associated with HPV infection and the progression of the disease [46,47]; on the other hand, *Gardnerella* was also found only in HPV-positive samples [49] and having a positive association with the presence of cancer [50]. 

The functional role of the microbiome in cervical cancer was also evaluated either by prediction from 16S rRNA or from shotgun metagenomics data [45,47,51,52]. Functional analysis found that the peptidoglycan synthesis pathway was upregulated in cervical cancer patients, folate biosynthesis and oxidative phosphorylation was upregulated in precancerous and cancer groups, whereas degradation of dioxin and 4-oxalocrotonate tautomerase and metabolism of starch and sucrose were downregulated [47,51]. Many other pathways, including biofilm formation, benzoate degradation, and others, were found to be enriched in healthy controls. In spite of this, PCA (Principal Component Analysis) could not distinguish between healthy and cancer patient samples based on their metabolic functions [51]. Moreover, cell motility pathways were negatively associated with cancer progression, while xenobiotic biodegradation and metabolism were positively associated with it [52].

Very few metatranscriptomic studies have been conducted on cervical cancer, and none of them examined the gene expression levels and associated functional profiles of cervical microbiomes. Ure et al. compared HPV-positive and HPV-negative cervical tumours and reported no significant differences in bacterial or viral genera between the two groups [53]. Arroyo Muhr et al. extracted DNA and RNA (with and without DNase I treatment) from the same samples and compared the resulting datasets. In general, RNA samples that were subjected to DNase treatment had significantly higher levels of microbial reads than DNAseq samples [50], indicating that such an approach leads to an improved recovery of the microbiome. Of note, analysing RNA allowed the detection of numerous viruses, including RNA viruses, in tumour and cervical swab samples, which could not be identified using metagenomics or 16S rRNA methods [50,53]. Moreover, it also led to the identification of HPV in patients who were HPV-negative based on PCR, confirming how the unbiased screening of RNA is superior to primer-based search [54].

## 5. Colorectal Cancer

One of the most extensively scrutinised diseases in terms of microbiome composition and function is colorectal cancer, which has been analysed using 16S rRNA profiling and metagenomic analysis, as well as by performing meta-analyses on previous metagenomic data. However, data based on metatranscriptomics are still scarce. In most studies, authors have collected faeces to profile microbiome composition and functions, although biopsy samples have also been used in some instances. Different CRC types, anatomical locations, and age-at-onset of the disease have been investigated in relation to the microbiome’s composition as well as how the microbiome impacts the response to treatment [55,56,57].

The two main findings stemming from CRC microbiome research are associated with the diversity of the microbial community and the enrichment and depletion of specific microbes. In particular, the faecal microbiome of colorectal cancer patients has a higher richness in comparison to healthy individuals, specifically with respect to species which are typically found in the oral flora [58]. In a meta-analysis of eight different metagenomic datasets, 29 bacterial species were specifically enriched in faecal samples from cancer patients. The bacteria included *Fusobacterium*, *Porphyromonas*, *Parvimonas*, *Peptostreptococcus*, *Gemella*, *Prevotella*, *Solobacterium*, *Dialister*, and *Clostridiales* [59]. Most of these genera are indeed oral commensal, and, in some cases, specific species such as *Parvimonas micra* and *Fusobacterium nucleatum* have been predicted to play a major role in CRC [60]. 

Some microbial species were identified as biomarkers, both geographically localised as well as global. For example, *F. nucleatum*, *Solobacterium moorei*, *Cenarchaeum symbiosum*, *Gemella morbillorum*, and *P. micra* were reported to be CRC biomarkers in more than one study, reinforcing the observation that an overabundance of oral commensal species in the gut is associated with the disease [58,61,62]. Other studies identified additional species, including *Ruminococcus torques*, *Porphyromonas asaccharolytica*, *Pasteurella stomatis*, and *Parvimonas* spp., as potential CRC biomarkers; however, these results have not yet been reproduced [58,59,61]. On the other hand, species including *Eubacterium eligens*, *Eubacterium ventriosum*, *Eubacterium hallii*, *Bifidobacterium catenlatum*, and *Gordonibacter pamelae* were found to be more abundant in controls [58,61,62]. Aside from bacteria, researchers have also screened for viruses and fungi in CRC microbiomes. They found that viruses are more prevalent in the gut microbiome of CRC patients relative to that of controls and that specific viral markers could be used to segregate diseased individuals [36]. Coker et al. profiled the mycobiome of colorectal cancer patients, and, similarly to the case of viruses, they reported that fungal members of the gut flora could be also used to discern patients from healthy individuals [63]. 

From a functional standpoint, cancer patients’ microbiomes revealed enrichment in genes involved in protein and mucin catabolism, higher levels of secondary bile acids, and higher levels of amino acids, cadaverine, and creatin [59,61]. It was also suggested that the microbes inhabiting CRC patients have enhanced glucogenesis, as well as putrefaction and fermentation pathways, while non-CRC microbes are associated with stachylose and starch degradation [58]. Another meta-study has predicted biomarkers primarily based on microbial function rather than taxonomic profiles, demonstrating that these biomarkers are superior to taxonomy-based biomarkers. Methylaspartate mutase sigma subunit activity; heptose II phosphotransferase activity; manganese, zinc, and iron transport system permease protein; and methyltransferase activity have all been shown to be related to colorectal cancer [60].

To date, seven metatranscriptomic studies have examined colorectal cancer, but not all of them quantified gene expression levels or derived active functions from them. Based on RNA-seq data, the enriched abundance of oral commensal in CRC microbiomes already recognised through 16S rRNA profiling and shotgun metagenomics has been confirmed. In particular, *F. nucleatum*, *P. micra*, and *Porphyromonas gingivalis* were the most prominent microorganisms, together with the gut-commensal *Bacteroides fragilis* [64,65]. An existing RNA-sequencing dataset was used to identify toxin genes, but the depth of coverage was insufficient to make a statistical analysis [66]. The most common form of hereditary CRC, Lynch syndrome, was also explored using 16S rRNA, metagenomics, and metatranscriptomics [67]. Using random forest (RF) prediction models, this study identified oxidative metabolic pathways as the major (albeit weak) predictors of disease progression. Some of those pathways have a protective effect, while others a deleterious one, highlighting how the microorganisms might mediate in one way or another the influence of oxidative stress in the progression of the disease. Very importantly, only those predictions based on transcript expression were significant, while neither taxonomy nor metagenomics-based functional analysis had any predictive power. Further, this study indicated that some of the least actively transcribed genes in the microbiome could be of low abundance, which is in line with a few other studies conducted on healthy individuals or cancer patients [68,69]. This indicates that the knowledge of which specific organism is expressing certain genes is extremely important. 

The role of microorganisms in manipulating oxidative stress in the CRC tumour microenvironment (TME) was confirmed by Lamaudiere et al. [70], who reported that pyruvate:ferredoxin/flavodoxin subsystems and H_2_O_2_ response are downregulated in CRC patients. On the other hand, mechanisms associated with positive selection, such as antibiotic resistance, host colonisation, biofilm formation, and horizontal gene transfer, were upregulated. At the level of specific transcripts, genes involved in the transport and uptake of vitamins and minerals, including iron, carnitine, selenium, and B-family vitamins, had their activity enhanced in CRC patients, which may contribute to often-observed deficiencies in patients. Finally, transcription patterns of specific genes could be associated with specific microbes: in particular, pathogenic bacteria showed enhanced transcriptional activity during cancerogenesis [69]. 

On the basis of eight different RNAseq datasets, cancer microbiomes were correlated with tumour microenvironment and immune cells. Most cancer-associated bacteria are positively correlated with NKC, whereas CD4+ T-cells, CD8+ T-cells, naive/pro B-cells, and T-regulatory cells are negatively correlated [71].

## 6. Gastric Cancer

The microbiome of gastric cancer patients and those with various levels of gastritis has been extensively investigated; however, the vast majority of the studies employed 16S rRNA sequencing while only one used metatranscriptomics [72]. Different studies on gastric cancer have also produced different results due to the different samples (gastric juices or biopsies), the anatomical location of the samples (proximal and distal stomach parts), and the progression of the disease (superficial gastritis and early or advanced cancer) [72,73,74,75,76].

A consistent finding across many studies is that, as the disease progresses, both the diversity and the richness of the microbial community of gastric cancer patients decline [73,74,77,78,79,80]. Also, it is well known that stomach cancer is strongly associated with *Helicobacter pylori* infection [81]. In particular, *Helicobacter* is enriched in precancerous and tumour-adjacent tissues, but its abundance declines in cancerous tissues, suggesting that the microbe might play a preparatory role in the onset of the disease [74,76,77]. As for other microbial species, their role in disease onset and progression is unclear, and several studies report contradictory results. On the one hand, *Propionibacterium* [77,78,79], *Prevotella* [74,76,78], *Streptococcus* [76,77,80], *Fusobacterium* [73,76,77,80], *Lactobacillus* [73,76,77,80], *Acinetobacter* [75,77], *Actinobacteria* [79], and *Atopobium* [76,80] have been found to be enriched in cancerous tissues; on the other hand, *Prevotella* and *Streptococcus* are reported in another study [79] to be underrepresented in the cancer cohort, and the depletion of *Actinobacteria* in cancer samples is replicated in several instances [73,75,76]. Interestingly, many gastric cancer-enriched bacteria are also found in the oral cavity of patients with superficial gastritis, including oral commensals *Parvimonas*, *Veillonella*, and *Peptostreptococcus* [74,80].

Looking at the functional landscape encoded by the gastric cancer microbiome (shown in Figure 4, together with the other cancer types) pathways associated with increased levels of small molecule synthesis and transport are typically active in the cancer microenvironment, such as peptidoglycan biosynthesis, nucleotide synthesis, transport and metabolism, amino acid transport and metabolism, and inorganic ion transport [74,77]. The virulence of *H. pylori* also plays a role in functional pathway activity [77], as well as the cancer stage: Park et al. report that typical hallmarks of malfunctioning replication, such as homologous recombination, mismatch repair, and DNA replication pathways, are overrepresented in advanced cancer samples [73]. Finally, a metabolomic analysis explored the influence of the anatomical location of cancer, associating amino acid metabolism-related pathways, such as arginine biosynthesis, protein digestion and absorption, alanine, aspartate, and glutamate metabolism, to distal cancer, while hormone-related pathways were enriched in proximal samples [76].

Thorell K et al., the only metatranscriptomic study, did not evaluate the functional potential of the microbiome, but detected *H. pylori* in people classified as *H. pylori* uninfected. In addition, they evaluated the expression pattern of *H. pylori* genes and found that pH regulation and nickel transport genes were highly expressed [72].

## 7. Oral Carcinoma

Oral cancers account for around 2–3% of all cancers and affect the lip, tongue, or mouth. Approximately 90% of diagnosed oral cancers are oral squamous cell carcinoma (OSCC). The five-year survival rate of oral cancer continues to hover at around 60% [82,83]. Various factors, such as smoking, alcohol, or genetic predisposition, are associated with OSCC. Interestingly, while the modern world saw a substantial decrease in smoking prevalence, this decrease did not lead to a decline in oral cancer occurrence [84,85]. Thus, other factors play a significant role in OSCC etiology, including carcinogenic or reactive oxygen species-producing microbiota [86].

Many studies used 16S rRNA sequencing to profile the microbiome in OSCC patients and compare it with a healthy cohort [85,87,88,89,90,91,92,93,94,95,96,97,98]. Often, the same bacteria were identified as a potential discriminant between healthy and non-healthy individuals. Bacteria belonging to phyla *Actinobacteria* [85,89,93,94,95,96] and *Firmicutes* [85,87,89,95,96] were consistently enriched in healthy individuals, with *Streptococcus* [85,87,89,90,91,92,93,95,96,97], *Rothia* [85,87,88,89,90,95,96,97,98], *Actinomyces* [85,88,89,93,95,97], *Veillonella* [87,88,94,95,97], and *Haemophillus* [90,93,94,98] representing the most overexpressed genera. Instead, in the microbiome of OSCC patients, genera *Fusobacterium* [87,88,90,91,92,93,95,96,97,98], *Campylobacter* [88,89,90,91,94,95,96,97], *Peptostreptococcus* [87,88,89,91,94,95,97,98], *Prevotella* [87,88,89,91,92,96], *Capnocytophaga* [89,91,95,96,97,98], and *Alloprevotella* [92,95,97,98] were predominantly enriched. Individual studies reported contrasting results, with *Actinomyces* [87], *Firmicutes* [88], and *Veillonella* [96] found depleted in healthy controls and *Prevotella* and *Capnocytophaga* depleted in OSCC patients [94]. *Fusobacterium*—the genus that was most consistently over-expressed in OSCC patients—is considered a normal element of the oral microbiome. Nevertheless, various virulence factors such as lipopolysaccharides (LPS), RadD (RecA-dependent accessory protein), or adhesins are produced by it. These factors have been associated with chronic inflammation, altered immune response, and cancer progression [99]. Furthermore, studies have shown that direct interaction between *F. nucleatum* and the epithelial cells plays a role in enhancing the progression of oral cancer [100]. *F. nucleatum*-induced inflammation has been shown to be impaired by *Streptococcus* species; hence, it is not surprising that a decline in the *Streptococcus* genus in OSCC patients is reported [101]. The enrichment of *Peptostreptococcus* in OSCC patients was also widely seen in microbiome profiling studies. It was shown that this genus upregulates the expression of TLR2 and TLR4 in CRC cells, favouring reactive oxygen species and enhancing cell proliferation [102]. 

Various functional pathways were predicted to be associated with OSCC from 16S rRNA data. While the majority of them were singular findings, some were identified in multiple studies. Notably, an increase in lipopolysaccharide biosynthesis [87,90,92,96,97], carbon fixation [88,93], and carbon metabolism [88,89] has been reported in OSCC patients.

Metagenomic studies reported a significantly increased abundance of cancer-related pathogen species *F. nucleatum* and a depletion of the genus *Streptococcus* and *Actinomyces* sp. *ICM47* in OSCC patients [87,103,104]. Functional differences were also analysed in these metagenomic studies. Two studies concordantly reported a decrease in nucleotide metabolism and amino acid metabolism in cancer patients [103,104]. Ganly et al. [103] also noted an increase in OSCC patients of several biosynthesis pathways, including vitamins, heme, sugars, and fatty acids (flavin, biotin, and thiamine). In contrast, folate biosynthesis, anaerobic energy metabolism, and pyruvate fermentation pathways were reported to be less active in OSCC patients. Liu et al. [104] instead focused on various virulence factors and identified a group of antimicrobial resistance genes (algR, flhF, ompA, lpxD, and ybtP) that were highly expressed by the species *F. nucleatum* and *P. endodontalis*, both of which were significantly more abundant in OSCC patients. 

Two metatranscriptomic studies on oral cancer patients were performed [105,106]. To advance early diagnostics for OC patients, Banavar et al. [105] employed machine learning techniques together with metatranscriptomics to develop a classifier that can accurately discriminate between oral cancer cases and healthy controls from saliva samples. The researchers noted a reduced abundance of genera *Streptococcus*, *Haemophilus*, and *Actinomyces* and high levels of genera *Fusobacterium* and *Prevotella* in the saliva of OSCC patients. Interestingly, a downward shift was observed where 75% of species and 81.6% of microbial functions were downregulated in cases compared to controls. Numerous functions, including biofilm formation, oxidative and non-oxidative metabolic pathways (microbial nitrate utilization, hydrogen sulphide production, and protein fermentation), and the production of carcinogenic metabolites such as benzaldehyde and arsenite, were identified among the strongest classifiers of cancer samples. Yost et al. [106] compared metatranscriptomic data between cancerous and tumour-adjacent tissues from OSCC patients and matched healthy tissue. The highest difference in taxonomic abundance was observed when comparing healthy patients with matched cancer tissues. Significantly overabundant microorganisms in cancer tissues were *Fusobacteria*, *Selenomonas* spp., *Capnocytophaga* spp., and genera of *Dialister* and *Johnsonella*, while the genus *Bacillus* and several other species were overrepresented in healthy microbiomes. Furthermore, the identification of differently expressed genes pointed to elevated levels of mineral transport (e.g., iron) and oxidative and non-oxidative functions, such tryptophanase and superoxide dismutase, in both tumour and tumour-adjacent samples when compared to the healthy controls. Finally, putative virulence factors expressed in the oral communities linked to OSCC indicated that activities related to the bacteria adaptation and survival, such as capsule biosynthesis, flagellum assembly, synthesis, and adhesion, were upregulated at tumour sites. *F. nucleatum* and *Fusobacterium periodonticum* were identified as the most active bacteria species expressing these virulence factors.

## 8. Lung Cancer

Among cancers, lung cancer has the highest mortality rate and is responsible for the majority of cancer-related deaths worldwide [107]. In contrast to previous assumptions, lungs are not sterile [108] and studies have shown that sterile mice or mice treated with antibiotics are significantly protected from lung cancer development caused by mutation of the p53 or KRAS genes [109], suggesting that the initial genetic trigger highly benefits from an infection-driven catalyst to fully degenerate into cancer. Therefore, the lung microbiome was studied to understand its role in carcinogenesis and response to treatment [110,111]. Several types of samples were analysed, including lung tumours and normal-adjacent tissue, bronchoalveolar lavages (BAL), sputum, saliva, and faeces [112,113,114,115,116,117,118,119]. Similar to all other cancers, most studies used 16S rRNA microbiome profiling and only a few utilised shotgun metagenomics or metatranscriptomics.

Based on the sample type (saliva, sputum, BAL, or tissue), the cancer type (small cell lung carcinoma (SCLC), adenocarcinoma (AC), or squamous cell carcinoma (SCC)) [120], the stage of the disease [121], and the anatomical location of the sample (upper vs. lower lobes) [118], the microbiome composition differs significantly. As a result of these differences, it is particularly difficult to describe lung microbiota comprehensively. Normally, the lung microbiota of healthy individuals are enriched with phyla *Firmicutes* and *Bacteroidetes* and members from genera *Prevotella*, *Veillonela*, and *Streptococcus* [108]. Despite using the same sample type, the results from different studies rarely converge, indicating the need for meta-studies, studies with greater sample sizes, or the integration of functional information. Microbiota analyses have found an association between lung cancer and the presence of *Streptococcus*, *Prevotella*, *Veillonella*, and *Capnocytophaga* in different types of samples [112,114,119,122,123,124]. Even though meta-analyses of data from various sample types found a similar increase in *Streptococcus* abundance, *Prevotella* abundance decreased in cancer patients [113]. A higher fraction of *Streptococcus*, *Prevotella*, or *Veillonella* was associated with a worse prognosis of survival [121]. Several other bacteria have also been found to be overabundant in cancer patients in comparison to non-cancer patients. In cancer patients’ BAL samples, *TM7*, *Gemminger*, *Blautia*, *Oscillapora*, *Ruminococcacea*, *Haemophilus*, *Fusobacterium*, *Neisseria*, and *Porphyromonas* have all been found to be enriched; however, contradictory data regarding the enrichment of these bacteria in cancerous samples have also been found [112,113,122,123]. As compared to normal-adjacent tissue, a meta-analysis of 16S rRNA data obtained from lung tissue showed a decrease in the abundance of phyla *Actinobacteria*, *Corynebacteriaceae*, and *Halomonadaceae* families, and genera *Corynebacterium*, *Lachnoanaerobaculum*, and *Holomonas* genera [125]. Some studies have also attempted to use the compositional characteristics of the microbiome as a predictive diagnostic tool to distinguish cancer patients from healthy individuals [112,123]. Evidence suggests that the type of cancer influences the composition of the lung microbiota [119]. In particular, a specific association was reported between *Acidovarax* and SCC patients, strongly mediated by smoking status [115], and between *Capnocytophaga* with AC patients [124]. Apart from bacteria, fungi might also play a role in the onset and progression of cancer, as reported by a recent mycobiome metagenomic study by Zhao et al. [126]. The authors reported that BAL samples from cancer patients possess a higher fungal diversity, and some fungal species are enriched compared to the controls. In particular, *Alternatia arborescens* seems to be the fungus most associated with the progression of the disease.

There have been very few attempts to evaluate the functional profile of lung microbiota. A metabolomics study was conducted on BALs and lung tissue flushings by Liu et al. Among lung cancer patients and controls, 40 metabolites showed significant differences. These metabolites were linked to 11 signalling pathways that were differently expressed in lung cancer patients, including apoptosis, autophagy, necroptosis, and sphingolipid signalling. An analysis of metagenomic data revealed differential enrichment between cancer patients and controls in five functional categories—K+ transporting ATPase, DNA polymerase III, PAS domain, membrane-associated protease RseP, and predicted flavoprotein YhiN [123]. Few studies have used 16S rRNA data to predict functional profiles of lung cancer microbes. Cancer patients, according to one of these studies, have enriched pathways related to ribosomes, pyrimidine metabolism, and purine metabolism, while pulmonary patients have enriched pathways related to two component systems, flagellar assembly and bacterial secretion [112]. Based on Huang’s study on different cancer stages, it has been found that pathways related to NAD salvage are significantly enriched in advanced cancer stages and are associated with *Granulicatella*, while an incomplete TCA reductive cycle is associated with *Peptostreptococcus* and L-valine biosynthesis is associated with *Pseudomonas* [116]. 

A third of the two main types of lung cancers (SC and NSC) involve a mutation in the *Egfr* gene, which codes for the epidermal growth factor receptor protein (EGFR) [127]. A few studies analysed the relationship between the mutational load of *Egfr* and the lung microbiome. Chang et al. [117] reported a negative correlation between the abundance of *Pseudomonas aeruginosa* in tumour tissue and a higher likelihood of mutations in the EGFR gene. A study based on 16S rRNA profiling found the degree of EGFR mutations and the abundance of genera *Bacteroidetes* and *Parvimonas* and *Actinobacillus* [116] positively correlated, while another did not identify a mediating role of the intratumoural bacterial burden on *Egfr* variants [128].

Only two studies claimed to take a metatranscriptomics approach in lung cancer research, yet their focus was on the correlation between the microbiome composition and the host’s, not the bacteria’s, transcriptional and functional response. The experiments from Chang et al. [117] pointed to the positive association of CD8+ T-cells, CD4+ naive T-cells, dendritic cells, and CD4+ central memory T-cells with patient survival. B-cells, on the other hand, were negatively correlated. Such high levels of immune cell types were also correlated with the presence of *Brevundimonas diminuta*, *Mycobacterium chelonae*, and *Mycobacterium franklinii* in tumour tissue. However, those results are far from being conclusive given that bacterial species were identified in only a small fraction of samples. Wong-Rolle and colleagues [128] employed spatial metatranscriptomics to show that tumour tissue has a higher bacterial burden than normal-adjacent tissue or tertiary lymphoid structures. Additionally, they found that cancer cells are a deeper reservoir of bacteria than immune cells. At first, this might seem surprising, considering that the natural role of immune cells is that of interacting with bacteria; however, immune cells eliminate bacteria, and these results suggest that cancer cells are instead better at adjusting their microenvironment to accommodate microbes that are beneficial to their survival and proliferation. Analysis of host gene expression revealed a positive correlation between intratumoural bacteria and Wnt/B-catenin, hypoxia, and angiogenesis pathways, while a negative correlation was found with genes involved in cell cycle (TP73) and pattern recognition (TLR5). This reinforces the hypothesis that, despite the immune system’s attempts to neutralise its activity, TME tries to shape its local microbiome and create favourable conditions for cancer growth. 

## 9. Melanoma

Melanoma, the most dangerous skin cancer, occurs when melanin-producing cells (melanocytes) undergo malignant mutations. It is estimated that in 2023, around 100 thousand new cases of melanoma will be diagnosed in the United states alone [24]. In the past decade, immunotherapies utilising immune checkpoint inhibitors (ICI), such as the anti-CTLA-4 antibody ipilimumab (IPI), anti-PD-1 antibodies nivolumab (NIVO), and pembrolizumab, became a new gold standard in treating advanced melanoma patients, substantially improving survival rate [129]. However, only around 50% of patients positively respond to ICI immunotherapies [130]. In order to understand whether the response to ICI is mediated by the microbiota composition, various meta-omics studies have been performed. 

Human health as a whole has been linked to gut microbiome alpha diversity, with decreased diversity being related to various acute and chronic diseases [131]. While most studies conducted so far have not reported an increase in the microbial community diversity within the ICI responders group [132,133,134,135], results based on 16S rRNA sequencing [136] and a combination of 16S and shotgun metagenomics [137] indicated that bacterial diversity in the gut microbiome can modulate the success of ICI treatment. When looking at specific microorganisms, some genera have been consistently reported in multiple studies to be overrepresented in the ICI responders group. These include *Faecalibacterium* [132,134,136,137], *Veillonella parvula* [135,138], *Ruminococcaceae* [132,137], *Streptococcus* [135,137], and *Coprococcus* [135,136]. Furthermore, members of the order *Bacteroidales* seem to have a detrimental effect on anti-cancer immune activation: they are negatively correlated with the density of the T-cell in pre-treatment tumours and the peripheral blood and associated with a reduction in peripheral cytokine response and an increase in immunosuppressive regulatory T-cells and myeloid-derived suppressor cells [137]. Other studies also identified a negative correlation between *Bacteroidales* order and *Bacteroides* genus prevalence and ICI treatment success, making it the most consistent negative marker to date [132,136]. As far as other bacteria are concerned, the evidence is so far contradictory. In some studies, ICI responders exhibited a higher abundance of *Ruminococcus gnavus*, *Bifidobacterium longum*, *Bacteroides thetaiotaomicron*, *Adlercreutzia equolifaciens*, and *Holdemania filiformis*, while non-responders showed enrichment of the same bacteria in other studies [132,134,135,136,138]. The observed variation may be attributed to a number of factors, such as geographical distribution, patients’ age, lifestyle, genetics, and variation in statistical approaches or sampling techniques. Moreover, the relatively low number of patients involved per group in each of these studies (N < 30) could amplify the effects of all the aforementioned variables.

A functional pathways prediction study from metagenomics shotgun sequencing (MGS) data by Gopalakrishnan et al. showed that anabolic functions including amino acid biosynthesis were predominant in responders while catabolic functions were more prevalent in non-responders [137]. Frankel et al. reported that responders’ microbiomes were enriched with bacterial enzymes involved in fatty acid synthesis. A strong positive correlation was also observed with inositol phosphate metabolism [134]. Wind et al. [135] identified 17 pathways that differed in abundance between responders and non-responders. Five major pathways in the responders group were aspartate superpathway, superpathway of thiamine diphosphate biosynthesis I, phosphorylation and dephosphorylation, superpathway of glycolysis, pyruvate dehydrogenase, TCA and glyoxylate bypass, and superpathway of thiamine diphosphate biosynthesis II. Only two pathways were identified in the non-responders group: namely, the peptidoglycan biosynthesis IV and the methanogenesis from H_2_ and CO_2_. Finally, in their study, Matson et al. failed to include any functional analysis of their MGS data [138]. A re-analysis of the data from the aforementioned study [132] indicated an increase in nucleotide biosynthesis among non-responders, whereas responders demonstrated elevated biosynthesis of complex organic compounds, such as isoprenoids, polyamines, and coenzymes.

Two metatranscriptomic studies were able to identify the contributors of specific pathways’ upregulation and downregulation. Peters et al. [136] observed that *Bacteroides* spp. were related to the adverse effects of the dysregulation of certain pathways. In particular, *B. ovatus* was associated with shorter progression-free survival via the upregulation of sugar (L-rhamnose) and vitamin B (pantothenate, pyridoxal 5-phospate, flavin, and 6-hydroxymethyl-dihydropterin diphosphate) biosynthesis pathways. Similarly, *B. dorei* and *B. massiliensis* were the major contributors to the upregulation of guanosine nucleotide biosynthesis pathways. Contrarily, petroselinate and synthesis pathways were related to a longer progression-free survival, with the latter being mediated by *Coprococcus eutactus*. A study by Usyk et al. [133] demonstrated a link between a patient’s microbiome and potentially life-threatening immune-related adverse events. The authors established a connection between adenosine metabolism and the increased occurrence of immune-related adverse events. The link between adenosine signalling and the suppression of tumour immunity has triggered the targeting of multiple signalling components in ongoing clinical trials [137]. 

Immunosuppressed individuals have a substantially augmented risk of developing non-melanoma skin cancer (NMSC), with the DNA of HPV commonly found in these tumours [139]. Arroyo Mühr et al. performed two separate studies [140,141] in which they investigated the link between HPV virus infections and the development of NMSC. In their first study, the researchers performed Illumina DNA sequencing on eight different NSCSs from a single patient that developed over a span of 10 years in comparison to eight different NMSCs from eight different patients. They concluded that viruses repeatedly found in independent tumours from a single patient might be more likely to play a role in tumorigenesis than viruses sporadically found in only a subset of specimens and acknowledged that the persistence of HPV over an extended period of time is an important factor for the maintenance of a tumour-conducive ME as well as for tumour progression. 

In a subsequent study, they analysed 345 NMSC samples developed after organ transplantation and identified that only 15/345 NMSCs were positive for HPV. The authors then concluded that HPV infections are likely unrelated to a significant increase in NMSC incidence after organ transplantation.

## 10. Conclusions and Final Outlook

The human microbiome plays an important role in the onset and progression of many cancers, as well as in the patient’s response to treatments, particularly in therapies targeting the immune system [142]. Up to now, most studies have focused on microbiome composition using 16S rRNA amplicon sequencing and shotgun metagenomics. Metatranscriptomics, on the other hand, has been seldom performed in cancer research. 

The identification of the microbiota composition is certainly important. Based on such information, it seems that the diversity of bacterial communities decreases during cancer development and that the physiology of many cancers benefits from high levels of *Fusobacteria* spp., especially *F. nucleatum*, in both the gut microbiome and the microbial environment specific to the tumour [28,32,33,34,58,60,61,87,103,106]. Moreover, other oral commensal species (e.g., *Prevotella* and *Parvimonas*) have been consistently reported to be overabundant in the gut microbiome or tumour tissues of cancer patients [59,74,75,78,87,89,92]. However, there is a great deal of inconsistency over the vast majority of species, indicating that microbial composition has limited power in discerning cancerous from healthy tissues. 

16S rRNA sequencing delivers no direct functional information, and even if there are methods to infer it (e.g., PICRUSt), the subsequent biological interpretation is extremely challenging. Compared to 16S rRNA sequencing, shotgun metagenomics is more powerful because it leads to the reconstruction of the entire genomes of all the microorganisms in a community. Data analysis can place fully assembled genomes more accurately on a phylogenetic tree, so that novel species can be identified. Also, information about whole genomes can be leveraged to predict the biological functions potentially active in a microbiome. This enables more consistent and reproducible findings since many different bacteria benefit from symbiosis with the cancer microenvironment through the same mechanisms. For example, shotgun metagenomics suggests that there are three main families of pathways that bacteria take advantage of in order to thrive in a tumour environment: oxidative stress, small molecule transport, and malfunctioning DNA repair [47,51,60,67,70,73,74,77,105,106]. Meta-analyses of aggregated metagenomics data could provide even more reliable results [59,61,62]. However, the mere presence of a microbe’s DNA does not indicate to what extent such a microbe is really active under certain circumstances, if at all. 

Metatranscriptomics captures all such information (Figure 5). A study of healthy adult men performed by Abu Ali et al. [68] reported that only 28% of the pathways potentially available based on metagenomics sequences were actually widely transcribed and considered as core metatranscriptomes. Basically, these are all housekeeping pathways, indicating that anything specific to a process and not required for mere survival would not be captured through metagenomics. Additionally, no quantitative conclusion could be drawn, since only one third of functional pathways were correlated with microorganism DNA and RNA abundance; finally, most functional pathways were enriched via the overexpression of genes only in specific members of the microbial community. Such a discrepancy was also observed between DNA abundance and RNA abundance of particular microbiota members in some cancer studies [70,141].

The other obvious added value of metatranscriptomics is the identification of bacteria that are actually responsible for the overexpression of certain genes. A study that analysed mice with gut inflammation, based on metagenomics data, could not find any statistically significant difference in pathways usage between the experimental and control groups. On the other hand, when analysing the metatranscriptomics data, they found immune-related and tissue damage functions to be highly upregulated through the specific activity of homeostasis-promoting bacteria, while *Proteobacteria* played a major role in increased pathogenicity [143]. In another case [106], the microbiome of oral carcinoma showed overexpression of virulence factors associated with activities related to bacteria adaptation and survival. Thanks to metatranscriptomics data, this could be attributed for the most part to *Fusobacterium* spp. Finally, in melanoma patients, *Bacteriodes* spp. and *Coprococcus eutactus* have been linked with the alteration of pathways leading to shorter and longer progression-free survival, respectively [136]. 

Importantly, there are situations in which the expression of the most biologically relevant genes comes from organisms that are not present in high abundance [68,70,143]. In these cases, the great importance of those bacteria would be missed without analysing the metatranscriptome. 

The predictive power of metatranscriptomics data is also extremely valuable. Poore et al. [144] detected specific microbial signatures in TCGA whole-genome and whole-transcriptome data from tissue and blood samples of most cancer types. Using data from blood tissues, the authors developed predictive models to differentiate cancer-free individuals from cancer patients with different types of cancer, as well as cancer patients at different stages. While all the data included had sufficient predictive power, RNA-sequencing data showed a higher level of robustness, in that the predictions’ accuracy was not affected by the specificity of the training dataset used in the model. Another study analysed microbial data from patients affected by CRC and Lynch syndrome (a precursor to CRC). They found that microbial transcriptional profiles were powerful predictors of LS-to-CRC progression, whereas taxonomic and metagenomics data were not [67].

Another advantage of metatranscriptomics is its much higher sensitivity in discovering whether a pathogen theoretically instrumental to cancer is actually an active commensal of the flora. For example, in a metatranscriptomic study, *H. pylori* was found to be active in the microbiome of both gastric cancer patients and healthy individuals initially considered to be *H. pylori* infection-free. This suggests that *H. pylori* may be a normal microbial community member of the stomach [72]. Similarly, another study found HPV infection in patients who had HPV-negative PCR results [54], and the authors remarked upon the power of an unbiased RNA-seq approach. 

Although it cannot technically be described as metatranscriptomics, the study of the interaction between the host molecular machinery (e.g., gene expression or protein levels, pathways activation, or immune response) and the composition of the microbiota also seems to be a promising approach, as demonstrated in breast cancer by the mediating role of *Listeria* spp. and *H. influenzae* on EMT and proliferative pathways [41] and that of *Gammaproteobacteria* and *Bacillus* on the depletion of ceramide [37]. Similarly, studies based on immune cells array [28,71,92] showed that a number of immune cells including NKC were enriched in breast, colorectal, and lung cancerous tissue, with specific bacteria being responsible for such upregulation, as well as the overexpression of particular pathways. 

Another feature unique to metatranscriptomics is the capability to identify RNA viruses, antisense RNAs, and small non-coding RNAs [145,146]. Moreover, since transcriptomic changes happen rapidly, repeated experiments at various time points could be used to determine antimicrobial efficacy as well as to provide deeper insight into the functional development of microbiota.

Finally, in the context of cancer research, the functional implications of the bacterial community are often more important than the discovery of novel species. Since metatranscriptomics requires, on average, 20 M reads per sample [147], against the 40–50 M reads recommended for shotgun metagenomics [148], one can sequence twice as many samples at the same cost, significantly increasing the detection power of the experiment.

Despite all the aforementioned advantages over metagenomics, a metatranscriptomics experiment has its own limitations and requires careful preparation. First, one needs to address the requirements of any experiment handling RNA. Sample collection and preservation are crucial due to the ease of degradation, as well as the appropriate choice of protocol for extraction and library preparation [149]. In particular, ribosomal depletion is recommended to minimise RNA contamination by host RNA, microbial ribosomal RNA, or transfer RNA. DNA contamination can also be an issue, and DNAse treatment is in fact also good practice in metatranscriptomics [50]. Specific to cancer is the need to obtain a rather large amount of starting material, given the little bacterial biomass present in tumour tissues relative to other samples, such as, e.g., faeces or saliva [150]. 

From a computational standpoint, there are a number of workflows to analyse metatranscriptomics data, many of which adapt tools that might not be specifically designed for the analysis of RNA from microbial communities [151]. In particular, researchers have not yet made the same efforts towards benchmarking and standardising analysis workflow from metatranscriptomics data as they did with metagenomics [152]: such efforts would certainly lead to increased accuracy, robustness, and reproducibility and allow for proper meta-analysis. This would, in turn, encourage researchers to more widely adopt metatranscriptomics and finally maximise its potential, which we believe is still largely untapped. 

## Figures and Tables

**Figure 1 ijms-24-13786-f001:**
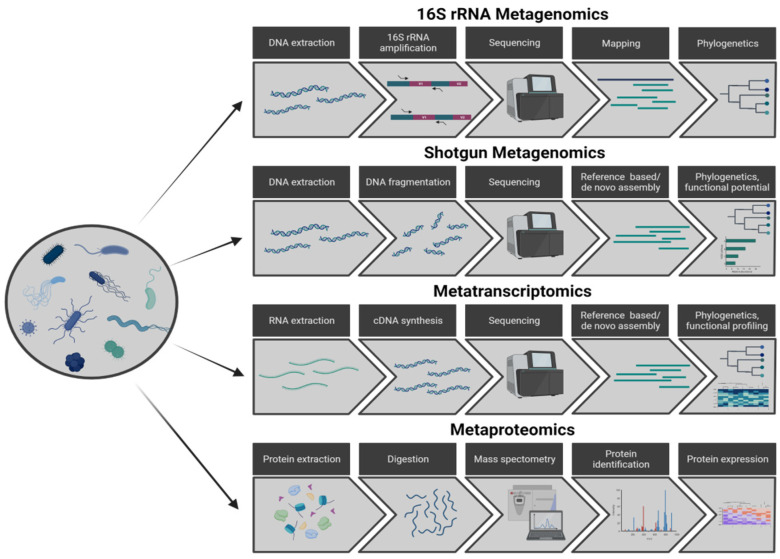
Graphic summary of the approaches to study the communities of microorganisms inhabiting an environment.

**Figure 2 ijms-24-13786-f002:**
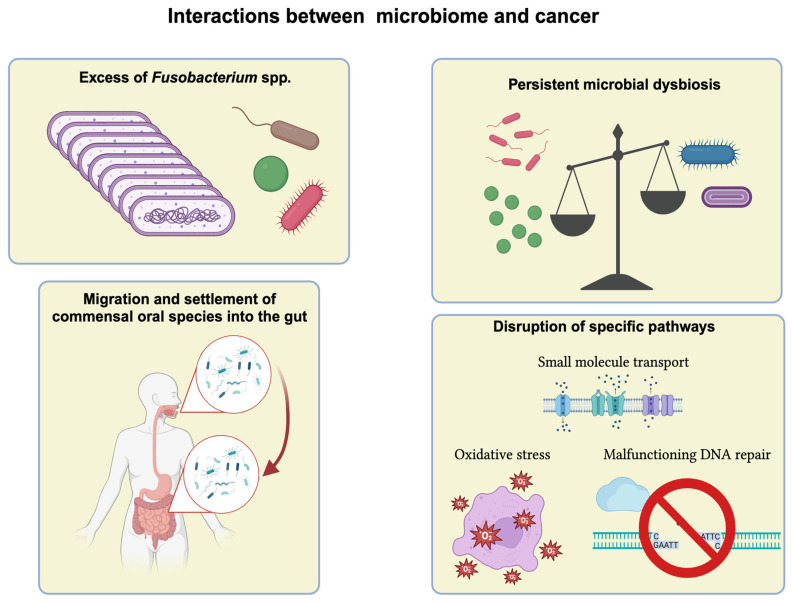
Summary of best characterised evidence to date on the interaction between microbiomes and cancer.

**Figure 3 ijms-24-13786-f003:**
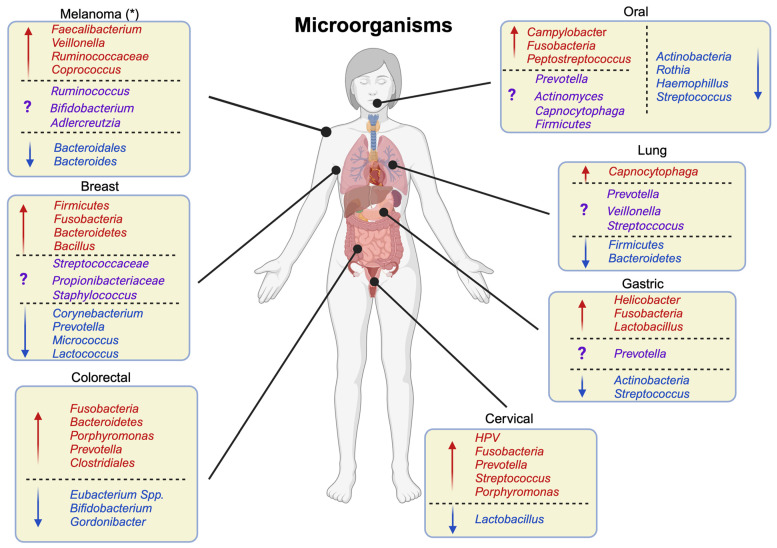
Overview of the microorganisms enriched (in red) or depleted (in blue) in multiple studies. The purple entries represent bacteria for which contrasting evidence has been reported. (*) The comparisons are between ICI respondents and non-respondents.

**Figure 4 ijms-24-13786-f004:**
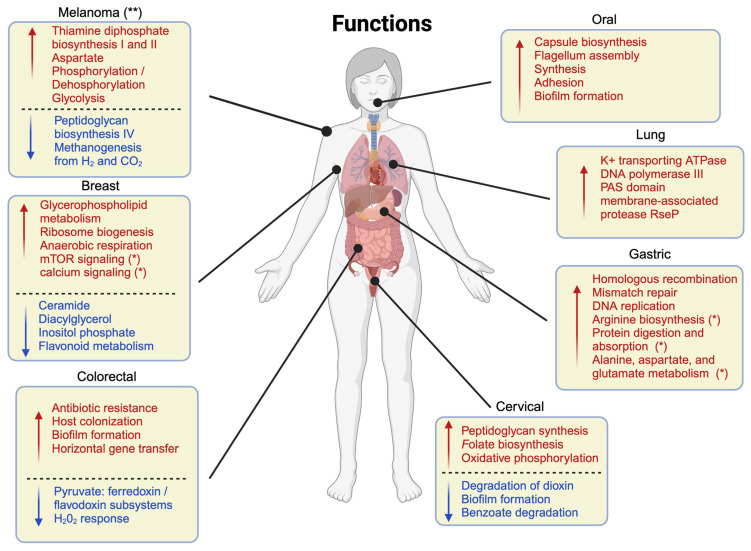
Overview of the functions and pathways enriched (in red) or depleted (in blue) in multiple studies. (*) The comparisons are between specific subsets of patients; refer to the text for details. (**) The comparisons are between ICI respondents and non-respondents.

**Figure 5 ijms-24-13786-f005:**
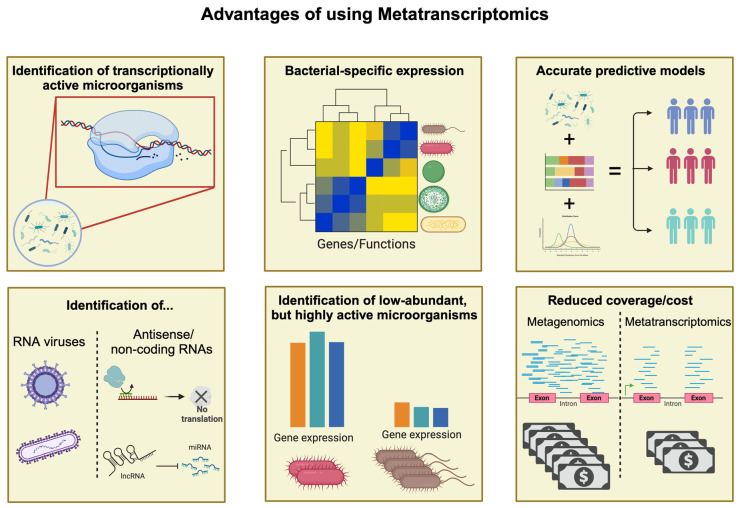
Additional benefits of using metatranscriptomics to study a microbial community.

**Table 1 ijms-24-13786-t001:** Summary of the studies analysed in this review. 16S: 16S rRNA metagenomics. MGx: shotgun metagenomics. MTx: metatranscriptomics.

Cancer Type	Papers (Total)	16S	MGx	MTx	MGx + MTx	16S + MTx	16S + MGx	16S + MTx + MGx	Other
Breast cancer	17	13	0	1	1	1	0	0	1
Cervical cancer	12	8	1	2	1	0	0	0	0
Colorectal cancer	17	2	6	6	0	0	1	2	0
Gastric cancer	9	8	0	1	0	0	0	0	1
Oral carcinoma	17	12	2	2	0	0	1	0	0
Lung cancer	18	11	2	1	0	2	1	0	1
Melanoma	7	0	4	0	1	0	1	1	0
TOTAL	97	54	15	13	3	3	4	3	3

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
