# Peer review of "Microbiomes, Their Function, and Cancer: How Metatranscriptomics Can Close the Knowledge Gap"

_ijms, 2023, doi:10.3390/ijms241813786_

Round 1

Reviewer 1 Report

Aitmanait et al. wrote a good review summarizing the connection between microbiome and cancer and how metatranscriptomics can benefit our understanding of the connection. This is a timely review that can guide people through the current studies of microbiome and cancer. The organization of the manuscript is good. The manuscript is divided into several sections based on the cancer types. Under each section, the studies were further grouped and summarized based on the microbiome detection methods they used. It will be a useful resource for researchers to learn more about the current study on the relations between cancer and microbiome. This review may encourage more people to use metatranscriptomics to study microbiome and the functional pathways considering its advantages. The following issues may be resolved to improve the presentation of this review.

Minor issues:

1.     Table 1: May add one row at the top or bottom to calculate the total papers for each method. This will make it clear that which method has been used a lot.

2.     The current version of the manuscript is too wordy with only one figure. It is very easy to get lost when reading through the manuscript. Including more figures may make it easier to explain the results of the reviewed studies. For example, a figure with a human body and all the cancer types highlighting the enriched microbes in different organs may be beneficial. Another figure with similar setup but highlighting the involved functional pathways instead of enriched microbes may also be useful.

Author Response

Many thanks to the reviewer for the positive feedback and suggestions. Here our responses: 

Table 1: May add one row at the top or bottom to calculate the total papers for each method. This will make it clear which method has been used a lot.
We have added the requested row. 

Including more figures may make it easier to explain the results of the reviewed studies.
We agree. We have added four figures: two along the lines suggested by the reviewer; one summarizing the strongest pieces of evidence overall; and one describing the advantages of using Metatranscriptomics. We believe this makes the manuscript more enjoyable and easier to read.

Reviewer 2 Report

This manuscript gave on overview on the applications of metagenomics and metatranscriptomics on the microbime-cancer research, and discussed the advantages of metatranscritptomics method in studying this type of study. I find overall this topic is interesting, however, there are a few major issues in this work that needed to be addressed.

Major comments:

1.     The review aims to give overview on the current metatranscriptomics applications on cancer-microbiome interactions, but the main text in fact has very limited metatranscriptomics studies, and mostly discussed about the metagenomics analysis on cancer-microbiome. The authors shall add more metatranscriptomics work in this review to fulfill their theme. For example, the authors mentioned very limited (1-2) studies in breast cancer, cervical cancer, gastric cancer, melanoma. And when they do, they mostly just list the studies without relating them to the topic and giving meaningful insights.

2.     The manuscript lacks of figures, which makes the review hard to read. The only figure is about technology workflow and not related with cancer. The authors could make summary figures to describe the relationship between cancer and microbiome to give reader background about this field. And the authors may also present the models from current studies to show how the metagenomics and metatranscriptomics advanced the cancer-microbiome field.

Minor comments:

1.     In the table, the cancer types shall have better naming. The cancer names should be full name consistently, for example, Breast Cancer instead of Breast. And the abbreviation shall have explanations in the table annotations.

2.     Under the “Summary of the studies reviewed”, the authors points that the largest number of papers analyses are CRC, gastric cancer and oral carcinoma, but it is not the case based on the numbers from Table 1, which should be Lung, Breast, CRC, and oral carcinoma.

Author Response

Thanks to the reviewer for the suggestions. 

Response to major comment #1:
we understand that the number of studies included is limited, but the problem cannot be overcome, since there are truly very few studies that used metatranscriptomics and cancer. This is the reason why we decided not to include only metatransciptomics papers, which was our initial idea. Pubmed returns: 

  • 3 papers with "Metatranscriptomic" or "Metatranscriptomics" + "Cancer" in the title
  •  22 papers in total with "Metatranscriptomic" or "Metatranscriptomics" in the Title or Abstract and "Cancer" in the title

Excluding reviews, these are 20 papers in total, which are all included in the manuscript.

Response to major comment #2:
We very much agree. We have added four figures:
Fig. 2 now shows the strongest evidence to date on the interaction between microbiomes and cancer.
Fig 3. shosw for each of the analysed cancers the microorganisms for which enrichment or depletion was confirmed by more than one study; also, the microorganisms with contrasting evidence are included. 
Fig. 4 is of  the same type as Fig. 3, but with the functions and pathways.
Fig. 5 summarizes all the advantages of Metatranscriptomics. 
We believe this makes the manuscript more enjoyable and easier to read.

Response to minor comment #1:
The naming in the table is now consistent and the abbreviations are explained in the caption. 

Response to minor comment #2:
Thanks. The sentence (first paragraph on Page 3) has been modified accordingly.  

Round 2

Reviewer 2 Report

The author's response has answered my suggestion. I agree with the publication of this article.